# Dietary 25 Hydroxyvitamin D3 Improved Serum Concentration Level and Alkaline Phosphatase Activity during Lactation but Had Meager Impact on Post-Farrowing Reproductive Performance in Sows

**DOI:** 10.3390/ani14030419

**Published:** 2024-01-27

**Authors:** Prester C. John Okafor, Nitipong Homwong

**Affiliations:** 1Laboratory of Swine Science, Department of Animal Science, Faculty of Agriculture at Kamphaeng Saen, Kasetsart University, Kamphaeng Saen Campus, Nakhon Pathom 73140, Thailand; presterchukajohn.o@ku.th; 2National Swine Research and Training Center, Faculty of Agriculture at Kamphaeng Saen, Kasetsart University, Kamphaeng Saen Campus, Nakhon Pathom 73140, Thailand

**Keywords:** vitamin D3, 25-hydroxyvitamin D3, alkaline phosphatase, epimers, post-farrowing, reproduction, performance, sow, lactation

## Abstract

**Simple Summary:**

Vitamin D3 regulates many biological functions in mammals. One of its classical roles is to maintain calcium homeostasis and improve bone metabolism. The dietary requirement of this vitamin in sows often exceed that required for finishing pigs due to their role in mitigating osteomalacia, also known as soft bone. Commercial variants of this vitamin also exist as 25 hydroxyvitamin D3; however, their function in the reproductive performance of sows is not clearly understood. In this study, two commercial products were compared with regular Vitamin D3 in feeds administered to sows from gestation to lactation. Post-farrowing reproductive performance, serum alkaline phosphatase activity, and 25 hydroxyvitamin D3 concentration were compared. Feed intake, pre-weaning mortality, and the number of weaned piglets differed during lactation. Alkaline phosphatase activity and 25 hydroxyvitamin D3 concentration increased during lactation. This could be caused by an increase in metabolic demand for phosphorus and calcium during lactation. The current finding shows that the use of 25 hydroxyvitamin D3 in sow diets may improve other functions such as bone strength, calcium, and phosphorus homeostasis without necessarily affecting sow reproductive performance.

**Abstract:**

Dietary 25 hydroxyvitamin D3 (25(OH)D3) promotes serum 25(OH)D3 concentration and alkaline phosphatase activity (ALP); however, post-farrowing reproductive performance of lactating sows fed with 14-epimer of 25(OH)D3 is uncertain. This study investigated post-farrowing reproductive performance, serum ALP activity, and serum 25(OH)D3 concentration in sows fed VD3, 25(OH)D3, or 14-epi 25(OH)D3. Weaned sows (n = 203) in parities 2 and 3 were blocked weekly and treated with 2000 IU/kg VD3 (T1), 25 μg/kg 25(OH)D3:14-epi 25(OH)D3 (T2), or 50 μg/kg 25(OH)D3 (T3) diets, all equilibrated to 2000 IU/kg as fed. Sow performance, treatment, and sampling period effects were analyzed. Environmental conditions were analyzed as covariates. The number of piglets weaned (*p* = 0.029), pre-weaning mortality (*p* = 0.029), sampling period (*p* < 0.001), and treatment and period interaction (*p* = 0.028) differed significantly. There was an increase in 25(OH)D3 during lactation due to physiological demands for milk calcium and milk production. Supplementing twice the concentration of 25(OH)D3 compared to its epimer, 25(OH)D3:14-epi 25(OH)D3, had no significant effect on the post-farrowing reproductive performance of lactating sows. The effect of 25(OH)D3 on post-farrowing reproductive performance and ALP activity in sows was influenced by metabolic demand for calcium due to physiological changes during lactation as well as epimer conformation.

## 1. Introduction

The need for improved reproductive performance has been a front-line objective in swine production [1,2]. Some crucial indices for estimating post-farrowing reproductive performance include litter size, live birth, the milking ability of sows, weaned piglets, pre-weaning mortality, and stillborn. These factors are also used in benchmarking [1,2]. The need for dietary support using nutrient supplements such as vitamin D3 (VD3) has also been extensively studied across various stages of sow reproductive life [3,4]. Metabolites of VD3 such as 25-hydroxyvitamin D3, 25(OH)D3, and 1,25-dihydroxyvitamin D3, 1,25(OH)2D3, are known to play a critical role during breeding, implantation, placentation, gestation, parturition, and lactation. For instance, the metabolism of available Ca produced during lactation largely depends on the regulatory functions of these metabolites [5]. They have also been reported to enhance milking ability, reduce stillbirth, and improve litter size [6,7,8]. A report by Weber and colleagues showed that the birth and weaning weights of piglets were improved [3]. The distinction in functions of VD3 and its metabolites in reproducing sows is still not clearly understood. Despite the above-mentioned effects of 25(OH)D3 on sow performance, some scholars reported no difference in the reproductive performance of sows fed either of these forms [9]. In addition to reproductive performance, metabolites of VD3 are well known regulators of alkaline phosphatase (ALP) activity.

The ALP enzymes are glycoproteins expressed during bone calcification, collagen formation, growth, and differentiation of tissues, as well as in bone remodeling [10]. This enzyme is highly expressed in growing animals due to an increase in the activity of osteoblasts. However, hepatic cells also play a role in augmenting the enzyme levels in mature animals [11]. The activity of ALP is required to metabolize inorganic phosphates, reduce pyrophosphates (which are known inhibitors of mineralization), and increase phosphate localization in osteons [12]. Reports have also shown that the activity of ALP is also promoted in a VD3 and 25(OH)D3 replete state [13,14]. Due to the crucial role of ALP in bone formation and mineralization, the present study aimed to understand the effects of the epimeric form of 25(OH)D3 on the activity of ALP in reproducing sows.

Epimers of 25(OH)D3 are analogues with similar structures and molecular weights as parent compounds, but they differ in the stereochemical structure of their respective side chains. The orientation of the compound often affects its biological function [15]. Some biologically active analogues include 3-epi 25(OH)D3 and 14-epi 25(OH)D3 produced by sigma-tropic hydrogen shifts at the third and fourteenth carbons of 25(OH)D3. This function is catalyzed by epimerases [16,17]. The method of synthesizing and harvesting these products might contribute to product differentiation [18]. Though chemical synthesis has been a common practice, fermentation technology has also been explored for commercial 25(OH)D3 production [19]. The 14-epi-analogs have been reported to promote the activation of the vitamin D receptor, a transcription factor that regulates genetic mechanisms involved in mineralization and bone fortification [20]. However, no studies have yet compared the 14-epimer of 25(OH)D3 in diets of gestating or lactating sows at peak parity. There is also a paucity of information on its role in ALP activity at this stage. The present study postulates that half a dose of 14-epimer of dietary 25(OH)D3 has a similar effect to the regular commercial 25(OH)D3 variant on serum concentration, ALP activity, and the reproductive performance of sows. Therefore, the objective of this study was to investigate post-farrowing reproductive performance, serum ALP activity, and 25(OH)D3 concentration in peak-parity sows fed dietary VD3 or either of two epimeric conformations of 25(OH)D3.

## 2. Materials and Methods

### 2.1. Animals and Dietary Treatments

A total of 203 weaned crossbreed sows (50% Landrace × 50% Yorkshire) in parity 2 and 3 (peak parity), which averaged 81 and 102 weeks of age, respectively, were included in the study. Sows were blocked by weaning week, and, on average, the study sample increased by approximately 11 sows for 18 consecutive weeks. Sows were randomly assigned to three treatments of approximately 67 sows each. Three dietary treatments contained either 2000 IU/kg of regular VD3 (T1), 25 μg/kg of 25(OH)D3:14-epi 25(OH)D3 (T2), or 50 μg/kg of 25(OH)D3 (T3). All treatment concentrations were equilibrated to 2000 IU/kg of VD3 in a base diet and supplied as mash feed for 20 weeks from gestation through lactation until weaning (Table 1). The nutrient compositions of gestation and lactation diets were formulated using FeedLIVE^®^ Version 1.61 (Live Informatics Co., Ltd., Nonthaburi, Thailand) and are shown in Table 2. Average feed intake was limited to 2.5 kg daily from day 1 to 84 (84 days) and was increased to 3.3 kg from days 85 to 109 (26 days) during gestation. During lactation, sows were fed ad libitum until weaned. Lactation length (LL, days) was managed by the producer, and hence was considered a covariate in this study. All suckling piglets were offered mash creep feed (VD3 composition, equivalent to 4000 IU/kg) from 10 days of age until weaning at approximately 25 days of age.

### 2.2. Feed Quality Control Analysis

As a quality control measure, five-hundred grams of feed from each batch supplied to the sows was sampled for proximate analysis. Dry matter (method 930.15), crude protein (method 2001.11), ether extract (method 2003.05), crude ash (method 942.05), crude fiber (method 978.10), calcium (method 927.02), and phosphorus (method 965.17) were analyzed according to the AOAC protocol [21]. The gross energy was determined using a bombs calorimeter (method ISO 9831) [22]. The outcome of the analysis is presented in Table 3. To quantify feed 25(OH)D3 concentration in treatment diets, feed samples were sent to BIOVET^®^ laboratory (Peshtera, Bulgaria). The result is presented in Table 4.

### 2.3. Environment, Housing, and Management

During gestation, sows were housed individually in stalls. From day 110, sows were moved to a farrowing barn. Individual farrowing stalls were equipped with heated crates and creep area. Sows and piglets had unlimited access to drinking water in their respective stalls by nipple. Additionally, water was supplied regularly in bowls for easy access to younger piglets. Where necessary, sows were assisted by injecting oxytocin during farrowing. Piglets were weighed within 24 h of farrowing. Cross-fostering to equalize litter size was carried out within 48 h after farrowing; however, this was restricted to sows farrowed within treatment groups. The gestation and lactation barns were equipped with evaporative cooling system.

### 2.4. Records and Measures

All performance records and inventory in the production facility were digitized in PigLIVE^®^ Version 4.0 (Live Informatics Co., Ltd., Nonthaburi, Thailand). A live record of the insemination date was documented; however, sows were confirmed in-pig 30 days after by a standing reflex during boar exposure. Gestation and lactation length (days) were days from insemination to farrowing and farrow to wean, respectively. Records collected at farrow were farrowing time (hours), which included the time from farrow of first piglet to completion, oxytocin use (ml), and percent oxytocin use. Post-farrowing reproductive performance indices recorded onsite were total born (TB), born alive (BA), still born, mummified, piglets and litter body weight (BW, kg), pre-weaning mortality, weaned piglets, and lactation feed intake (LFI, kg/day). Sow body condition, mainly BW and backfat (BF, mm) before farrowing, and at weaning, as well as percent BW and BF loss during lactation were also measured. Fecal score was recorded daily by observing piglets’ fecal droplets in each pen. Stools were assigned 0, 1, or 2 indicating lumpy (no diarrhea), pasty, or liquid (diarrhea) stool, respectively.

### 2.5. Blood Sample Collection and Biochemical Analysis

Blood samples were collected by periods on day 5 post-farrowing (AF5), 25 post-farrowing (AF25), and day 6 post-weaning (AW6). One piglet from each sow was sampled for blood collection at weaning. Approximately 3 mL of blood was obtained from each sow through the jugular vein, aptly transferred into coagulant blood collection tubes, and transported in an ice box to the Laboratory of Swine Science, Faculty of Agriculture at Kamphaeng Saen for processing. Centrifugation was carried out at 2500× *g* for 15 min at 4 °C and the resulting serum was collected into 1.5 mL microtubes and stored at −20 °C for further analyses of 25(OH)D3 concentration (ng/mL) and ALP activity (U/L).

### 2.6. Alkaline Phosphatase Assay

The in vitro test for the quantification of serum ALP activity was carried out using an ALP kit (Mindray^®^, Shenzhen, China). The reagents consist of the following: R1, magnesium acetate-zinc sulfate in a solution of 2-amino-2-methyl-1-propanol buffer (2.5, 1.2 and 435 mmol/L respectively); and R2, p-nitrophenyl phosphate (60 mmol/L). A multi sera calibrator and control were used for quality control assessment (reference range: 80.6–98.6 U/L). Distilled water was used as a blank. The reaction volume for samples and reagents (sample: R2:R1) was set to 1:12.5:50. The absorbance reading was obtained using a chemistry analyzer (Mindray BS-120^®^, Shenzhen, China) at a reaction temperature of 37 °C and wavelengths between 405 and 546 nm. The detection range for linearity was between 5 and 800 U/L. The equation for the reaction can be stated as follows:$$4-\mathrm{Nitrophenylphosphate}+H_{2}O+\mathrm{ALP}+{\mathrm{Mg}}^{+}\to \;4-\mathrm{Nitrophenol}+P_{i}$$

ALP catalyzes the hydrolysis of 4-nitrophenyl phosphate, producing 4-nitrophenol and inorganic phosphate. The alkaline buffer also acts as a phosphate-group acceptor. The activity of ALP is directly proportional to the rate of formation of 4-nitrophenol in the sample.

### 2.7. Analysis of Serum 25(OH)D3 Concentration

Sample preparation: Serum samples and acetonitrile were added dropwise into a microtube at a ratio of 1:2 (*v*/*v* serum/acetonitrile) and mixed thoroughly via vortex. The preparation was centrifuged thrice at 5000× *g* for 10 min at 25 °C, and the supernatant carefully collected in clean microtubes, filtered through a nylon membrane (0.22 μm, Whatman^®^, Kent, UK) into an amber glass vial, and transferred to an autosampler for chromatographic analysis.

Chromatographic analysis: Reverse-phase symmetry C18 column, (5 µM 4.6 × 250 mm, Waters, San Ramon, CA, USA) was used in this study. The mobile phase contained 100% acetonitrile, delivered at a flow rate of 1.2 mL/min. Sample injection volumes were set at 50 μL. Injector and column temperatures were set at 25 and 40 °C, respectively. Chromatographic separation occurred at a detection wavelength of 264 nm using a photodiode array (PDA) detector (Waters, San Ramon, CA, USA).

Calibration curve: Standards of VD3 and 25(OH)D3 (Ehrenstorfer^®^ GmbH, Augsburg, Germany) were used to generate calibration curves. Using the mobile phase as a diluent, 128, 64, 32, 16, 8, and 4 ng/mL of VD3 and 25(OH)D3 were prepared and used as calibrators. The chromatogram is presented in Figure 1.

### 2.8. Statistical Analysis

Gestation or lactation in days were analyzed using general linear models according to the statistical model: Y_ijk_ = µ + W_j_ + T_k_ + e_ijk_. The term, “µ” is the coefficient for grand mean; Y_ijk_ is the gestation or lactation length of i^th^ sow in k^th^ treatment inseminated in a specific week j^th^. Residual error in the model was denoted as e_ijk_.

Post-farrowing reproductive performance was analyzed using general linear models. A statistical model was considered as follows: Y_ijkl_ = µ + W_j_ + LL_i_ +T_k_ + e_ijkl_, where Y_ijkl_ is the observed reproductive performance variables of sow_i_ with lactation length (LL_i_) in the treatment group (T_k_), from a lactation sow in weaned week j^th^ (W_j_) with the experimental error, e_ijkl_. Lactation length (LL_j_) was considered as the model covariate when needed.

Piglet fecal scores from sows in treatments were analyzed using a non-parametric Kruskal–Wallis test.

The general linear mixed-effects model for an analysis of 25(OH)D3 concentration and ALP activity was as follows: Y_ijk_ = µ + T_j_ + P_k_ + T_j_:P_k_ + SOW_i_ + e_ijk_, where Y_ijk_ = vector of response for i^th^ sow in j^th^ treatment at period k^th^ of blood collection; µ = grand mean; SOW_i_ was the random effect; T_j_ was treatment effects; P_k_ was period effects; T_j_:P_k_ was treatment–period interaction effects; and the term e is the coefficient for the residual error of all model terms.

Residual distributions from general linear models and general linear mixed-effects models were studied. Assumptions for normality, linearity, and heteroskedasticity were tested. In cases of non-normality, Box–Cox transformation was used to estimate transformation parameters [23]. Arcsine square root transformation was also applied in modes when deemed appropriate. Model selection was performed using Akaike Information Criteria (AIC) for either nested or un-nested models. The lower the AIC, the better the model fit. Graphs were plotted using “ggplot 2” package [24], and treatment means were compared using the “emmeans” package [25]. All statistical analyses were carried out using R version 4.2 [26]. Statistical significance was set at *p* < 0.05.

## 3. Results

### 3.1. Post-Farrowing Reproductive Performance

The effect of dietary supplementation of 25(OH)D3 on post-farrowing reproductive performance is presented in Table 5. At a 95 percent confidence level, no significant difference was observed in gestation length; however, lactation length differed significantly between T1 and T2 (*p* = 0.039). Lactation length was designated as a covariate for all subsequent statistical models. Lactation feed intake also differed significantly between sows T2 and T3 (*p* = 0.023). Figure 2 shows the average daily feed intake during lactation. Neither of the three treatments showed a significant difference in body weight nor backfat within this period of lactation. Pre-weaning mortality as well as the number of piglets weaned were significantly different between treatments. The average pre-weaning mortality of piglets varied significantly between T1 and T2 (*p* = 0.029). Weaned piglets were also statistically significant between treatments (*p* = 0.029). The average fecal score for all treatment groups during lactation was less than two, indicating no severe sign of diarrhea.

### 3.2. ALP

The activity of ALP was estimated on three levels, treatment, period, and interaction (treatment × period) (Table 6). The activity of ALP did not differ significantly at the treatment level. There was a significant difference in ALP activity at AF5, AF25, and AW6 (*p* < 0.001). The interaction between treatment and period was also statistically different (*p* = 0.028). At AF25, ALP activity was at its peak with estimated values of 100.80 ± 10.52, 77.56 ± 10.52, and 96.70 ± 10.87 U/L in T1, T2, and T3, respectively. The lowest estimated activity was recorded at AW6 (Figure 3A).

### 3.3. 25(OH)D3

The concentration of 25(OH)D3 was also established at the treatment, period, and interaction levels. As shown in Table 6, there was no significant difference in treatment means; however, the period effect was shown to be statistically significant (*p* < 0.001). The interaction model plotted in Figure 3B showed that 25(OH)D3 concentration was at its peak at AF25 with estimated values of 55.78 ± 11.83, 80.86 ± 10.59, and 80.02 ± 10.53 ng/mL for T1, T2, and T3, respectively.

The cocktail used for standard preparation contained 50 ng/mL dry mass of VD3 and 25(OH)D3. Separation was carried out in a reverse phase symmetry C18 column. 25(OH)D3 had a shorter retention time compared to VD3.

Feed intake increased non-linearly during lactation, with an average of 6.37, 6.11, and 6.58 in T1, T2, and T3, respectively. The average lactation length was 25 days. Treatments 1 (VD3), 2 (25(OH)D3:14-epi 25(OH)D3), and 3 (25(OH)D3) are diets fed to the experimental groups and their respective VD3 or 25(OH)D3 compositions.

Blood samples were collected at three periods: AF5—5 days post-farrowing; AF25—25 days post farrowing; and AW6—6 days post-weaning. There was an increase in alkaline phosphatase activity at AF25 in all treatments. Similarly, 25(OH)D3 concentration was also higher at AF25, however, with higher peaks in treatments 2 and 3 compared to the control. Treatments 1 (VD3), 2 (25(OH)D3:14-epi 25(OH)D3), and 3 (25(OH)D3) are diets fed to the experimental groups and their respective VD3 or 25(OH)D3 compositions.

## 4. Discussion

### 4.1. Post-Farrowing Reproductive Performance

To the best knowledge of the authors, this is the first study evaluating the effects of supplementing dietary 25(OH)D3 and its 14-epimer on reproductive performance, ALP activity, and serum 25(OH)D3 concentration in multiparous sows. Since the 14-epi-analogue of 25(OH)D3 purportedly has a higher potency than the regular form, a half-dose was used to provide the right empirical stance to substantiate or rescind the claim. The chromatographic analysis of feed samples showed a two-fold higher dietary concentration of 25(OH)D3 compared to 14-epi-25(OH)D3; however, this did not result in a two-fold increase in post-farrowing reproductive performance, indicating that the relationship between dietary concentration and reproductive performance may not be linear. Lactation feed intake, pre-weaning mortality, and the number of piglets weaned differed significantly between treatments. Because lactation length was strictly managed by the farm, it was considered as a covariate for subsequent analyses. The lengthy lactation culminated in a linear increase in feed intake. It was also shown that higher feed intake during lactation was associated with positive body reserve and improved estrus cycling [27]. This has a positive impact since lengthy lactation often resulted in improved milk production in sows, and, in turn, improved piglet vitality [28]. In other studies, sows with longer lactation length had a low wean-to-estrus interval in days [29,30]. Therefore, the practice in itself is intended to improve overall sow and piglet performance. The mortality rate reported herein was lower compared to that published previously [31]. This could be attributed to differences in farm management practices. According to Muns et al. (2016), pre-weaning mortality can be caused by the piglet, sow, or environment [32]. All housing facilities used for gestation and lactation sows in the present study were equipped with an evaporative cooling system, annulling the possibility of heat-induced mortality. A plausible explanation for the observed mortality rate in sows fed 14-epi 25(OH)D3 could be low piglet vitality and low feed intake, which are often associated with milking ability. The scope of this study did not incorporate the study of milking ability in sows; hence, empirical evidence was provided in this regard. Other studies have, however, demonstrated that piglet mortality and consequent low weaning proportion in sows fed 25(OH)D3 diets were linked to reduced milking [3,33]. With regard to feed intake, it is a known fact that sows consume more feed to replenish body reserve, which tends to derogate during lactation; therefore, adequate milking is often associated with reduced body reserve (often shown by reduction in backfat). In the current study, 14-epi 25(OH)D3-fed sows not only had the lowest feed intake during lactation, but comparatively had the lowest reduction in back fat. This could possibly result in low milking, low piglet vitality, and mortality. The current discussion so far has shown a notable distinction in lactation feed intake between the experimental groups. This finding agrees with a previous report stating that dietary 25(OH)D3 did not improve sow reproductive performance more than regular VD3 [9]. Both feed intake and the average number of weaned piglets differed significantly between experimental groups. These factors were mostly influenced by farm management practice and might not lead to any significant production loss or pejorative effect.

### 4.2. ALP

After an adjustment for treatment effects, the activity of ALP differed significantly with respect to the sampled periods. Higher activity was observed at AF25 compared to AF5 and AW6. The current observation further supports the need to consider the physiological stage of sows in studies involving ALP. Similar findings have also been reported in other studies [34,35]. The need for phosphorus (P) is the most probable driving factor for increased ALP activity during lactation. At the jejunum of the small intestine, dietary P is absorbed into the bloodstream and transported to organs where it mediates various metabolic functions, including those of fetal and mammary gland development during gestation and lactation, thereby contributing significantly to the total P level in the circulation [11]. In a P replete state, unbound P can be readily absorbed into the circulation; this process drives the activity of ALP [36]. Calcitriol, a hormonal form of VD3, is also one of the essential regulators of P homeostasis. A possible mechanism was explained in Kozai et al. where it was shown that 1-alpha hydroxylase is often expressed abundantly in a state of phosphate depletion [37]. Consequently, the depletion of phosphate will lead to an increased metabolic demand for P and, in turn, an increase in ALP activity.

### 4.3. 25(OH)D3

There was a significant difference in mean 25(OH)D3 concentrations at sampled periods, the highest being at AF25. The linear increase in dietary feed intake during lactation suggests that the observed changes in 25(OH)D3 concentration might partly be influenced by dietary levels. The current observation consents the finding that post-farrowing concentration increased linearly with dietary feed levels [38,39]. However, no statistical difference was observed at the treatment level. This shows that higher dietary concentration alone might not necessarily culminate in increased bioavailability. Hence, several biological factors could jointly affect serum concentration. As seen in previous reports, evidence has also shown that blood concentration is not only influenced by dietary levels but metabolic Ca demand as well [40,41]. Studies have shown that during lactation, available Ca and P are often elevated. The observed increase in the concentration of 25(OH)D3 at AF25 indicates an upsurge in the activity of 1-alpha hydroxylase, possibly in response to metabolic Ca and P demand [42]. Consistent with the current findings, the activities of 1-alpha hydroxylase and ALP are co-regulated in response to dietary Ca and P levels, which are the hormonal triggers for the assembly of their respective transcription machinery [35,42]. Though epimerization causes variation in the biological activity of 25(OH)D3, serum level is to a greater extent influenced by physiological changes as well as sows’ nutrient demand during lactation.

## 5. Conclusions

The current study lucidly demonstrated the distinction in function of two 25(OH)D3 products differing in their respective epimeric conformations. The comparatively low lactation feed intake in 14-epi 25(OH)D3-fed sows led to a reduction in the average number of weaned piglets and slightly higher mortality. These results are a culmination of factors that are closely linked and can be improved by good farm management practices. Due to their subjective nature, their influence on the outcome of reproductive performance in this study can be considered meager. The current finding therefore supports the study hypothesis that half doses of 25(OH)D3:14-epi 25(OH)D3 and 25(OH)D3 have a similar effect on sows’ reproductive performance. The finding that neither the activity of ALP nor 25(OH)D3 concentration were influenced by dietary treatments also assents the current null hypothesis. However, the major distinction observed at AF25 implies that bioavailable concentrations of 25(OH)D3 were also influenced by physiological changes as well as the metabolic demand for Ca and P, as seen during lactation. The 25(OH)D3:14-epi 25(OH)D3 is therefore a highly potent substitute for the regular 25(OH)D3 and VD3 variants in reproducing sows.

## Figures and Tables

**Figure 1 animals-14-00419-f001:**
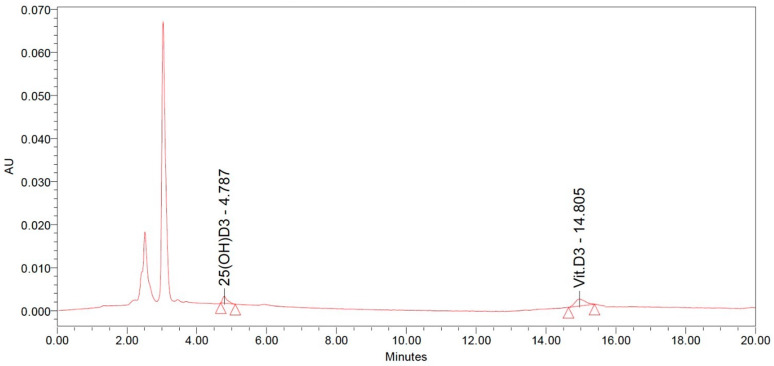
Chromatogram of vitamin D3 and 25-hydroxyvitamin D3 extraction using 100 percent acetonitrile.

**Figure 2 animals-14-00419-f002:**
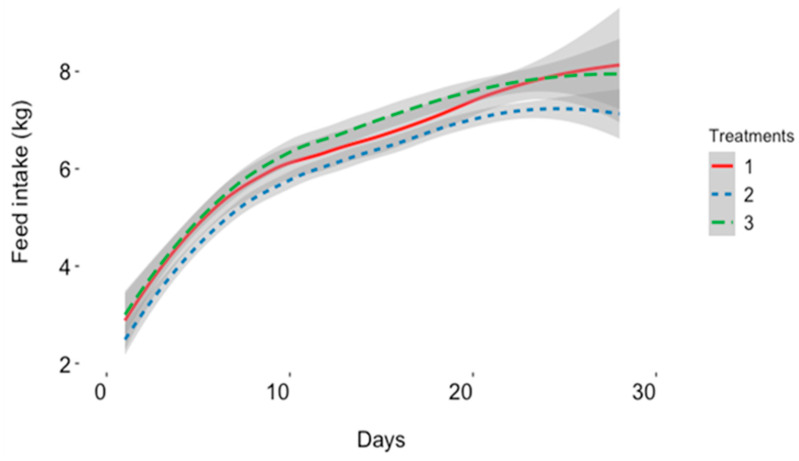
Estimate of average daily feed intake of multiparous sows during lactation.

**Figure 3 animals-14-00419-f003:**
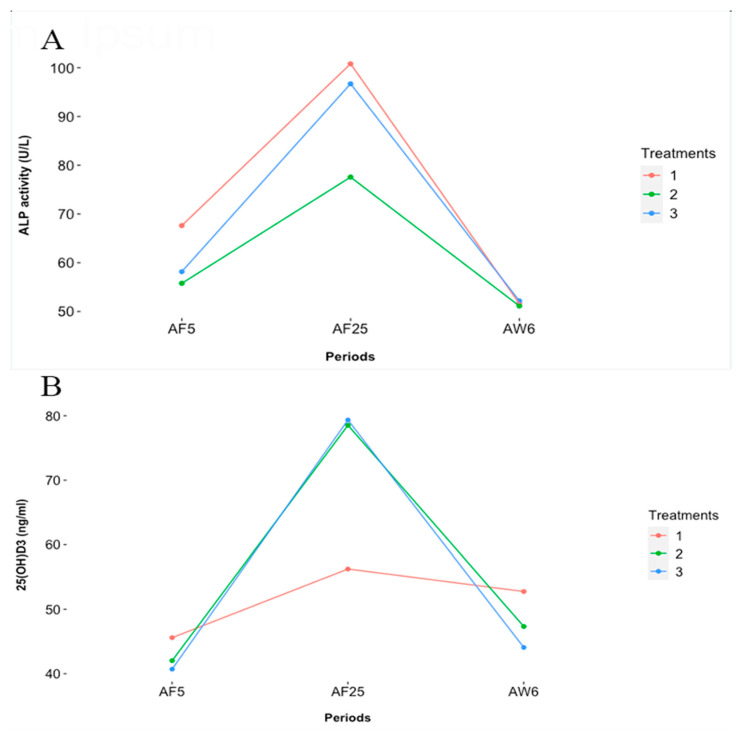
Interaction plots (treatment × sampled periods) depicting (**A**), alkaline phosphatase activity and (**B**), 25(OH)D3 concentration in lactating sows.

**Table 1 animals-14-00419-t001:** Forms and concentrations of vitamin D3 used as a dietary supplement.

Treatments	Active Substance/Product	Mass (g)/Metric Ton of Feed	Dose per kg of Diet	Product Conc. IU/Metric Ton
T1 ^a/^	500,000	4	2000 IU	2,000,000
T2 ^b/^	69.7 mg	360	25 μg/kg	2,000,000
T3 ^b/^	12.5 g	4	50 μg/kg	2,000,000

T1 = VD3 (1 mg of VD3 = 500 IU); T2 = 25(OH)D3:14-epi 25(OH)D3 (1 mg = 80,000 IU); T3 = 25(OH)D3 (1 mg = 40,000 IU); ^a/^ Regular form of cholecalciferol, VD3; ^b/^ Metabolite form of cholecalciferol: 25-hydroxycholecalciferol, 25(OH)D3.

**Table 2 animals-14-00419-t002:** Feed ingredients and calculated nutrient composition of basal diets for gestating and lactating sows.

Ingredient, %	Gestating Sow	Lactating Sow
Broken rice	10.00	35.00
Tapioca meal (70%)	30.00	5.00
Rice barn	23.29	15.00
Wheat barn	15.00	12.00
Soybean oil	1.99	5.70
Soybean meal (45.5%)	15.78	22.44
L-lysine	0.15	0.65
DL-methionine	0.08	0.13
L-threonine	0.07	0.09
Monodicalcium phosphate	1.08	1.41
Calcium carbonate	1.79	1.80
Salt	0.25	0.27
Premix ^1^	0.50	0.50
^¶^ Optiphose^®^	0.01	0.01
Calculated nutrient composition, %		
Metabolizable energy, MJ/kg	12.34	13.81
Crude protein	14.00	17.50
Crude fat	6.22	8.79
Crude fiber	5.97	4.71
Calcium	1.15	1.15
Total phosphorus	0.86	0.85
Available phosphorus	0.41	0.45
Sodium	0.32	0.32
Lysine	0.80	1.36
Methionine	0.27	0.38
Methionine + Cystine	0.48	0.63
Threonine	0.53	0.68
Tryptophan	0.17	0.21

^1^ Dietary premix per kilograms of feed contented as following; without vitamin D_3_, vitamin A 3000 IU, vitamin E 11 IU, vitamin K_3_ 1.00 mg, vitamin B_1_ 0.40 mg, vitamin B_2_ 1.20 mg, vitamin B_6_ 1.50 mg, vitamin B_12_ 0.01 mg, pantothenic acid 4.00 mg, niacin 4.00 mg, folic acid 0.30 mg, biotin 0.40 mg, choline chloride 60.00 mg, ferrous 40.00 mg, copper chelate 36 mg, manganese chelate 10.80 mg, zinc chelate 36.00 mg, cobalt 0.40 mg, iodine 0.40 mg, and selenium 0.06 mg. ^¶^ phytase enzyme used in diets.

**Table 3 animals-14-00419-t003:** Proximate analysis showing the respective treatment composition of various nutrients in diets.

Diet	Gestating Sow			Lactating Sow		
Treatment	T1 ^‡^	T2 ^‡^	T3 ^‡^	T1 ^‡^	T2 ^‡^	T3 ^‡^
Gross energy ^1^, MJ/kg	18.77	19.31	19.40	18.02	17.67	17.61
Crude protein, %	15.30	15.37	15.09	19.49	18.79	18.93
Ether extract, %	7.78	8.26	7.85	12.54	11.78	12.23
Crude fiber, %	4.32	4.67	4.31	3.91	3.46	3.98
Crude ash, %	7.46	7.07	7.12	7.50	7.19	7.11
Calcium ^2^, %	1.29	1.22	1.18	1.39	1.35	1.27
Phosphorus, %	0.81	0.81	0.80	0.96	0.89	0.87

^1^ Gross energy was analyzed using a bombs calorimeter (method ISO 9831). ^2^ Calcium was analyzed by using Atomic Absorption Spectrophotometry (Shimadzu, AA-7000). ^‡^ T1: Basal diet with Vitamin D_3_ 2000 IU plus Optiphos^®^ 0.1 g per kg diet. T2: Basal diet with 25 μg 14-epi 25(OH)D_3_ plus Optiphos^®^ 0.1 g per kg diet. T3: Basal diet with 50 μg 25(OH)D_3_ per plus Optiphos^®^ 0.1 g per kg diet.

**Table 4 animals-14-00419-t004:** Outcome of quality control analysis for 25(OH)D3 level in gestating and lactating diets.

Diet	Level (ng/g)		
	T1	T2	T3
Gestation	<2.00	20.20	57.30
Lactation	<2.00	22.70	50.00

Dietary 25(OH)D3 assay was analyzed using HPLC by BIOVET^®^ laboratory (Peshtera, Bulgaria); the limits of detection and limits of quantitation were 2 and 5 ng/g at 10% coefficient of variation. T1 (VD3), T2 (25(OH)D3:14-epi 25(OH)D3), and T3 (25(OH)D3) represent the experimental groups and their designated dietary VD3 or 25(OH)D3 compositions.

**Table 5 animals-14-00419-t005:** Post-farrowing reproductive performance of sows and piglets.

Parameter	T1	T2	T3	*p*-Value
	Mean ± SEM			
Gestation length, days	117.93 ± 0.20	118.04 ± 0.20	118.41 ± 0.20	^1/^ 0.168
Lactation length, days	23.74 ± 0.21 ^b^	24.49 ± 0.22 ^a^	24.30 ± 0.22 ^ab^	^1/^ 0.039 *
LFI, kg/day	6.37 ± 0.15 ^ab^	6.11 ± 0.14 ^b^	6.58 ± 0.14 ^a^	^1/^ 0.023 *
BW before farrowing, kg	297.60 ± 2.79	298.25 ± 2.77	299.46 ± 2.85	^1/^ 0.895
BW at weaning, kg	257.23 ± 3.15	252.89 ± 3.26	260.59 ± 3.27	^1/^ 0.250
BF before farrowing, kg	18.35 ± 0.25	18.80 ± 0.25	18.65 ± 0.25	^1/^ 0.414
BF at weaning, mm	17.78 ± 0.24	17.33 ± 0.25	17.65 ± 0.25	^1/^ 0.421
BW loss, % ^¶^	14.40 ± 0.57	14.84 ± 0.60	13.50 ± 0.60	^1/^ 0.104
BF loss, % ^¶^	10.32 ± 1.28	11.99 ± 1.33	8.37 ± 1.34	^1/^ 0.156
Total born, head	15.73 ± 0.38	16.45 ± 0.37	15.72± 0.38	^1/^ 0.278
Piglets BW, kg	1.47 ± 0.03	1.40 ± 0.03	1.41 ± 0.03	^1/^ 0.140
Piglets born alive, head	14.15 ± 0.36	14.94 ± 0.35	14.67 ± 0.36	^1/^ 0.286
Stillborn piglets, head	1.61 ± 0.15	1.39 ± 0.15	1.13 ± 0.16	^1/^ 0.100
Mummified piglets, head	0.48 ± 0.09	0.66 ± 0.09	0.69 ± 0.09	^1/^ 0.228
Pre-wean mortality, head	0.81 ± 0.13 ^a^	1.26 ± 0.13 ^b^	0.84 ± 0.13 ^ab^	^1/^ 0.029 *
Weaned piglets, head	11.42 ± 0.13 ^a^	10.97 ± 0.13 ^b^	11.39 ± 0.13 ^a^	^1/^ 0.029 *
Time of farrowing, hour	4.39 ± 0.28	4.15 ± 0.25	4.16 ± 0.31	^1/^ 0.086
Oxytocin dose, ml	0.86 ± 0.13	1.05 ± 0.13	1.13 ± 0.14	^1/^ 0.294
Litter fecal score, score	1.44 ± 0.13	1.70± 0.13	1.41 ± 0.13	^2/^ 0.399

^1/^ *p*-value obtained from a general linear model. ^2/^ *p*-value obtained from non-parametric test using Kruskal–Wallis test. ^¶^ Arcsine square root transformation applied to response variable. ^ab^ Different superscripts within the same row indicate statistical significance (*p* < 0.05). * *p* < 0.05. LFI, lactation feed intake; SEM, standard error of mean; BW, body weight; BF, back fat. Litter fecal score was as follows: 0 (no diarrhea/solid); 1 (pasty but not liquid); 2 (diarrhea/liquid). T1 (VD3), T2 (25(OH)D3:14-epi 25(OH)D3), and T3 (25(OH)D3) fed sows.

**Table 6 animals-14-00419-t006:** Effects of treatment and periods on ALP activity and 25(OH)D3 concentration in lactating sows.

Treatment Effects				
Parameters	T1	T2	T3	*p*-Value
	Mean ± SEM			
ALP, U/L ^‡^	73.35 ± 10.02	61.49 ± 10.02	69.02 ± 10.36	^1/^ 0.820
ALP AF5, U/L	67.62 ± 6.81	55.78 ± 6.81	58.17 ± 7.04	^2/^ 0.527
ALP AF25, U/L	100.82 ± 10.86	77.58 ± 10.86	96.66 ± 11.24	^2/^ 0.865
ALP AW6, U/L	51.65 ± 4.70	51.15 ± 4.70	52.16 ± 4.86	^2/^ 0.805
25(OH)D3, ng/mL ^‡^	54.67 ± 9.37	61.96 ± 9.01	58.20 ± 8.98	^1/^ 0.880
25(OH)D3, AF5, ng/mL	54.90 ± 7.84	49.69 ± 7.09	40.23 ± 7.52	^2/^ 0.724
25(OH)D3, AF25, ng/mL	55.78 ± 11.83	80.86 ± 10.59	80.02 ± 10.53	^2/^ 0.219
25(OH)D3, AW6, ng/mL	60.62 ± 9.14	48.55 ± 8.86	52.09 ± 7.97	^2/^ 0.637
Period effects				
	AF5	AF25	AW6	
ALP, U/L	60.58 ± 6.14 ^a^	91.58 ± 6.14 ^b^	51.64 ± 6.14 ^c^	^1/^ <0.001 ***
25(OH)D3, ng/mL	42.59 ± 5.76 ^a^	72.33 ± 5.43 ^b^	47.75 ± 5.73 ^a^	^1/^ <0.001 ***
Treatment × Period				
ALP, U/L ^†^	—	—	—	^1/^ 0.028 *
25(OH)D3, ng/mL ^†^	—	—	—	^1/^ 0.146

^1/^ *p*-value obtained from general linear mixed-effect model. ^2/^ *p*-value obtained from general linear model. ^‡^ Total treatment effect adjusted for individual sow difference and sample collection periods. ^†^ Interaction plots shown in Figure 3. ^abc^ Different superscript within the same row indicate statistical significance (*p* < 0.05). * *p* < 0.05, *** *p* < 0.001. LFI, lactation feed intake; SEM, standard error of mean; BW, body weight; BF, back fat. T1 (VD3), T2 (25(OH)D3:14-epi 25(OH)D3), and T3 (25(OH)D3) fed sows.

## Data Availability

The data are available on request from the corresponding author.

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
