# Peer review of "Dietary 25 Hydroxyvitamin D3 Improved Serum Concentration Level and Alkaline Phosphatase Activity during Lactation but Had Meager Impact on Post-Farrowing Reproductive Performance in Sows"

_animals, 2024, doi:10.3390/ani14030419_

Round 1
Reviewer 1 Report
Comments and Suggestions for Authors
The work is interesting and generally well written, but it requires some corrections and explanations.
Please complete the research hypothesis.
Please provide information on the exact chromatograph on which vit. D3 determinations were performed.
A description of the abbreviations of the experimental groups should be provided under each Table.
l.207-208 - The difference is only between the T1 and T2 groups, please correct the description.
l.211-212 - A difference is shown between the T2 and T3 groups.
Please let me know how the mixtures with the addition of vitamin D3 were prepared so that they were homogeneous, because the addition of vitamin D3 was very small.
Please explain what Optiphose is.
Table 2 - why were the basic components of the experimental mixtures not analyzed, and were they based only on calculated values?
Please complete Fig. 1 with a description of the groups (T1, T2, T3).
Please correct Conclusions. What is presented in this chapter is not a conclusion but a summary of the results. You need to write down what results from the research conducted.
Please prepare References in accordance with the Journal's requirements (instructions for Authors)!
Comments on the Quality of English Language-
Reviewer 2 Report
Comments and Suggestions for Authors
The work is interesting and may be the subject of publication in Animals, but only after completing the indicated comments and suggestions.
Line 86 - Please provide your research hypothesis before the research objective. We are the research hypothesis that … or Our research hypothesis was that …
Line 94 - Were the studies carried out in a research or production pig facility?
Line 110-111 - Provide analytical procedure numbers (AOAC, 2012) for dry matter, crude protein, ether extract, crude ash, crude fiber, calcium and phosphorus.
Line 135 - Provide information regarding piglet fecal scores.
Line 141 - Provide the blood collection procedure, e.g. where the blood was collected from (what vein), how long after feeding, whether an anticoagulant was used, etc.
Lines 212-213; 220-223 and others. There is no need to repeat the results tabulated. Just describe the trends and highlight the statistical significance of the differences.
Table 1. How was the value of metabolic energy calculated? Provide the formula and reference, convert to MJ EM. For individual nutrients, add "crude", e.g. Crude protein, etc.
Table 3. How was the gross energy content in feed determined? Provide the procedure and reference, convert to MJ EM. Provide this information on line 112.
Line 350 - In the Conclusions, it would be worth mentioning what suggestions regarding the use of various forms of vitamin D result from the research conducted? What recommendations for practical pig feeding?
